# INFERENCE SUBOPTIMALITY
# IN VARIATIONAL AUTOENCODERS

## ABSTRACT

Amortized inference has led to efficient approximate inference for large datasets. The quality of posterior inference is largely determined by two factors: a) the ability of the variational distribution to model the true posterior and b) the capacity of the recognition network to generalize inference over all datapoints. We analyze approximate inference in variational autoencoders in terms of these factors. We find that suboptimal inference is often due to amortizing inference rather than the limited complexity of the approximating distribution. We show that this is due partly to the generator learning to accommodate the choice of approximation. Furthermore, we show that the parameters used to increase the expressiveness of the approximation play a role in generalizing inference rather than simply improving the complexity of the approximation.

## 1 INTRODUCTION

There has been significant work on improving inference in variational autoencoders (VAEs) (Kingma & Welling, 2014; Rezende et al., 2014) through the development of expressive approximate posteriors (Rezende & Mohamed, 2015; Kingma et al., 2016; Ranganath et al., 2016; Tomczak & Welling, 2016; 2017). These works have shown that with more expressive approximate posteriors, the model learns a better distribution over the data.

In this paper, we analyze inference suboptimality in VAEs: the mismatch between the true and approximate posterior. In other words, we are interested in understanding what factors cause the gap between the marginal log-likelihood and the evidence lower bound (ELBO). We refer to this as the inference gap. Moreover, we break down the inference gap into two components: the *approximation gap* and the *amortization gap*. The approximation gap comes from the inability of the approximate distribution family to exactly match the true posterior. The amortization gap refers to the difference caused by amortizing the variational parameters over the entire training set, instead of optimizing for each datapoint independently. We refer the reader to Table 1 for detailed definitions and Figure 1 for a simple illustration of the gaps. In Figure 1, $\mathcal{L}[q]$ refers to the ELBO using an amortized distribution $q$, whereas $q^*$ is the optimal $q$ within its variational family.

Our experiments investigate how the choice of encoder, posterior approximation, decoder, and model optimization affect the approximation and amortization gaps. We train VAE models in a number of settings on the MNIST, Fashion-MNIST (Xiao et al., 2017), and CIFAR-10 datasets.

Our contributions are: a) we investigate inference suboptimality in terms of the approximation and amortization gaps, providing insight to guide future improvements in VAE inference, b) we quantitatively demonstrate that the learned true posterior accommodates the choice of approximation, and c) we demonstrate that using parameterized functions to improve the expressiveness of the approximation plays a large role in reducing error caused by amortization.

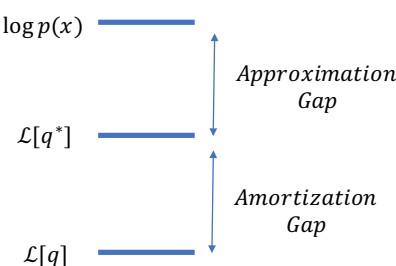

Figure 1: Gaps in Inference

| Term | Definition | VAE Formulation |
|---|---|---|
| Inference | $\log p(x) - \mathcal{L}[q]$ | $\text{KL}\left(q(z|x)||p(z|x)\right)$ |
| Approximation | $\log p(x) - \mathcal{L}[q^*]$ | $\text{KL}\left(q^*(z|x)||p(z|x)\right)$ |
| Amortization | $\mathcal{L}[q^*] - \mathcal{L}[q]$ | $\text{KL}\left(q(z|x)||p(z|x)\right) - \text{KL}\left(q^*(z|x)||p(z|x)\right)$ |

Table 1: Summary of Gap Terms. The middle column refers to the general case where our variational objective is a lower bound on the marginal log-likelihood (not necessarily the ELBO). The right most column demonstrates the specific case in VAEs. $q^*(z|x)$ refers to the optimal approximation within a family $\mathcal{Q}$, i.e. $q^*(z|x) = \arg\min_{q \in \mathcal{Q}} \text{KL}\left(q(z|x)||p(z|x)\right)$.

## 2 BACKGROUND

### 2.1 INFERENCE IN VARIATIONAL AUTOENCODERS

Let $x$ be the observed variable, $z$ the latent variable, and $p(x, z)$ be their joint distribution. Given a dataset $X = \{x_1, x_2, ..., x_N\}$, we would like to maximize the marginal log-likelihood:

$$\log p(X) = \sum_{i=1}^{N} \log p(x_i) = \sum_{i=1}^{N} \log \int p(x_i, z_i) dz_i. \tag{1}$$

In practice, the marginal log-likelihood is computationally intractable due to the integration over the latent variable $z$. Instead, VAEs optimize the ELBO of the marginal log-likelihood (Kingma & Welling, 2014; Rezende et al., 2014):

$$\log p(x) = \mathbb{E}_{z \sim q(z|x)}\left[\log\left(\frac{p(x, z)}{q(z|x)}\right)\right] + \text{KL}\left(q(z|x)||p(z|x)\right) \tag{2}$$

$$\geq \mathbb{E}_{z \sim q(z|x)}\left[\log\left(\frac{p(x, z)}{q(z|x)}\right)\right] = \mathcal{L}_{\text{VAE}}[q]. \tag{3}$$

From the above we can see that the lower bound is tight if $q(z|x) = p(z|x)$. The choice of $q(z|x)$ is often a factorized Gaussian distribution for its simplicity and efficiency. VAEs perform amortized inference by utilizing a recognition network (encoder), resulting in efficient approximate inference for large datasets. The overall model is trained by stochastically optimizing the ELBO using the *reparametrization trick* (Kingma & Welling, 2014).

### 2.2 EXPRESSIVE APPROXIMATE POSTERIORS

There are a number of strategies for increasing the expressiveness of approximate posteriors, going beyond the original factorized-Gaussian. We briefly summarize normalizing flows and auxiliary variables.

#### 2.2.1 NORMALIZING FLOWS

Normalizing flow (Rezende & Mohamed, 2015) is a change of variables procedure for constructing complex distributions by transforming probability densities through a series of invertible mappings. Specifically, if we transform a random variable $z_0$ with distribution $q_0(z)$, the resulting random variable $z_T = T(z_0)$ has a distribution:

$$q_T(z_T) = q_0(z_0)\left|\det\frac{\partial z_T}{\partial z_0}\right|^{-1} \tag{4}$$

By successively applying these transformations, we can build arbitrarily complex distributions. Stacking these transformations remains tractable due to the determinant being decomposable: $\det(AB) = \det(A)\det(B)$. An important property of these transformations is that we can take expectations with respect to the transformed density $q_T(z_T)$ without explicitly knowing its formula known as the law of the unconscious statistician (LOTUS):

$$\mathbb{E}_{q_T}[h(z_T)] = \mathbb{E}_{q_0}[h(f_T(f_{T-1}(...f_1(z_0))))] \tag{5}$$

Using the change of variable and LOTUS, the lower bound can be written as:

$$\log p(x) \geq \mathbb{E}_{z_0 \sim q_0(z|x)} \left[ \log \left( \frac{p(x, z_T)}{q_0(z_0|x) \prod_{t=1}^{T} \left| \det \frac{\partial z_t}{\partial z_{t-1}} \right|^{-1}} \right) \right]. \tag{6}$$

The main constraint on these transformations is that the determinant of their Jacobian needs to be easily computable.

### 2.2.2 AUXILIARY VARIABLES

Deep generative models can be extended with auxiliary variables which leave the generative model unchanged but make the variational distribution more expressive. Just as hierarchical Bayesian models induce dependencies between data, hierarchical variational models can induce dependencies between latent variables. The addition of the auxiliary variable changes the lower bound to:

$$\log p(x) \geq \mathbb{E}_{z,v \sim q(z,v|x)} \left[ \log \left( \frac{p(x, z) r(v|x, z)}{q(z, v|x)} \right) \right] \tag{7}$$

$$= \mathbb{E}_{q(z|x)} \left[ \log \left( \frac{p(x, z)}{q(z|x)} \right) - \mathrm{KL}\Big( q(v|z, x) \| r(v|x, z) \Big) \right] \tag{8}$$

where $r(v|x, z)$ is called the reverse model. From Eqn. 8, we see that this bound is looser than the regular ELBO, however the extra flexibility provided by the auxiliary variable can result in a higher lower bound. This idea has been employed in works such as auxiliary deep generative models (ADGM, Maaløe et al. (2016)), hierarchical variational models (HVM, Ranganath et al. (2016)) and Hamiltonian variational inference (HVI, Salimans et al. (2015)).

### 2.3 MARGINAL LOG-LIKELIHOOD ESTIMATION

We use two bounds to estimate the marginal log-likelihood of a model: IWAE (Burda et al., 2016) and AIS (Neal, 2001). Here we describe the IWAE bound. See Section 6.5 in the appendix for a description of AIS.

The IWAE bound is a tighter lower bound than the VAE bound. More specifically, if we take multiple samples from the $q$ distribution, we can compute a tighter lower bound on the marginal log-likelihood:

$$\log p(x) \geq \mathbb{E}_{z_1 \ldots z_k \sim q(z|x)} \left[ \log \left( \frac{1}{k} \sum_{i=1}^{k} \frac{p(x, z_i)}{q(z_i|x)} \right) \right] = \mathcal{L}_{\mathrm{IWAE}}[q]. \tag{9}$$

As the number of importance samples approaches infinity, the bound approaches the marginal log-likelihood. This importance weighted bound was introduced along with the Importance Weighted Autoencoder (Burda et al., 2016), thus we refer to it as the IWAE bound. It is often used as an evaluation metric for generative models (Burda et al., 2016; Kingma et al., 2016). As shown by Bachman & Precup (2015) and Cremer et al. (2017), the IWAE bound can be seen as using the VAE bound but with an importance weighted $q$ distribution.

## 3 METHODS

### 3.1 APPROXIMATION AND AMORTIZATION GAPS

The inference gap $\mathcal{G}$ is the difference between the marginal log-likelihood $\log p(x)$ and a lower bound $\mathcal{L}[q]$. Given the distribution in the family that maximizes the bound, $q^*(z|x) = \arg\max_{q \in \mathcal{Q}} \mathcal{L}[q]$, the inference gap decomposes as the sum of approximation and amortization gaps:

$$\mathcal{G} = \log p(x) - \mathcal{L}[q] = \underbrace{\log p(x) - \mathcal{L}[q^*]}_{\text{Approximation}} + \underbrace{\mathcal{L}[q^*] - \mathcal{L}[q]}_{\text{Amortization}}. \tag{10}$$

For VAEs, we can translate the gaps to KL divergences by rearranging (2):

$$\mathcal{G}_{\text{VAE}} = \underbrace{\text{KL}\big(q^*(z|x)||p(z|x)\big)}_{\text{Approximation}} + \underbrace{\text{KL}\big(q(z|x)||p(z|x)\big) - \text{KL}\big(q^*(z|x)||p(z|x)\big)}_{\text{Amortization}}. \qquad (11)$$

### 3.2 FLEXIBLE APPROXIMATE POSTERIOR

Our experimentation compares two families of approximate posteriors: the fully-factorized Gaussian (FFG) and a flexible flow (Flow). Our choice of flow is a combination of the Real NVP (Dinh et al., 2017) and auxiliary variables (Ranganath et al., 2016; Maaløe et al., 2016). Our model also resembles leap-frog dynamics applied in Hamiltonian Monte Carlo (HMC, Neal et al. (2011)).

Let $z \in \mathbb{R}^n$ be the variable of interest and $v \in \mathbb{R}^n$ the auxiliary variable. Each flow step involves:

$$v' = v \circ \sigma_1(z) + \mu_1(z) \qquad (12)$$
$$z' = z \circ \sigma_2(v') + \mu_2(v') \qquad (13)$$

where $\sigma_1, \sigma_2, \mu_1, \mu_2 : \mathbb{R}^n \to \mathbb{R}^n$ are differentiable mappings parameterized by neural nets and $\circ$ takes the Hadamard or element-wise product. The determinant of the combined transformation's Jacobian, $|\det(Df)|$, can be easily evaluated. See section 6.2 in the Appendix for a detailed derivation.

Thus, we can jointly train the generative and flow-based inference model by optimizing the bound:

$$\log p(x) \geq \mathbb{E}_{z,v \sim q(z,v|x)} \left[ \log \left( \frac{p(x,z')r(v'|x,z')}{q(z,v|x) |\det(Df)|^{-1}} \right) \right] = \mathcal{L}_{\text{flow}}[q]. \qquad (14)$$

Additionally, multiple such type of transformations can be stacked to improve expressiveness. We refer readers to section 6.1.2 in the Appendix for details of our flow configuration adopted in the experimentation.

### 3.3 EVALUATION BOUNDS

We use several bounds to compute the inference gaps. To estimate the marginal log-likelihood, $\log \hat{p}(x)$, we take the maximum of our tightest lower bounds, specifically the maximum between the IWAE and AIS bounds. To compute the AIS bound, we use 100 chains, each with 500 intermediate distributions, where each transition consists of one HMC trajectory with 10 leapfrog steps. The initial distribution for AIS is the prior, so that it is encoder-independent.

For our experiments, we test two different variational distributions: the fully-factorized Gaussian $q_{FFG}$ and the flexible approximation $q_{Flow}$ as described in section 3.2. When computing $\mathcal{L}_{\text{VAE}}[q]$ and $\mathcal{L}_{\text{IWAE}}[q]$, we use 5000 samples. To compute $\mathcal{L}_{\text{VAE}}[q^*]$, we optimize the parameters of the variational distribution for every datapoint. See Section 6.4 for details of the local optimization and stopping criteria.

## 4 RELATED WORK

Much of the earlier work on variational inference focused on optimizing the variational parameters locally for each datapoint, e.g. the original Stochastic Variational Inference scheme (SVI, Hoffman et al. (2013)) specifies the variational parameters to be optimized locally in the inner loop. Salakhutdinov & Larochelle (2010) perform such local optimization when learning deep Boltzmann machines. More recent work has applied this idea to improve approximate inference in directed Belief networks (Hjelm et al., 2015).

Most relevant to our work is the recent work of Krishnan et al. (2017). They explicitly remark on two sources of error in variational learning with inference networks, and propose to optimize approximate inference locally from an initialization output by the inference network. They show improved training on high-dimensional, sparse data with the hybrid method, claiming that local optimization reduces the negative effects of random initialization in the inference network early on in training. Yet, their work only dwells on reducing the amortization gap and does analyze the error arising from the use of limited approximating distributions.

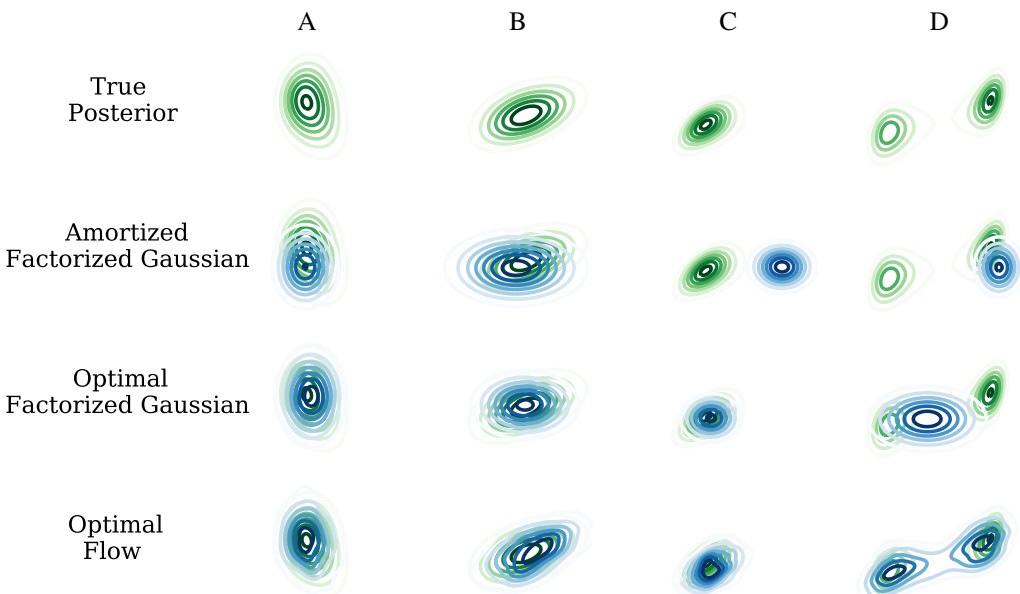

Figure 2: True Posterior and Approximate Distributions of a VAE with 2D latent space. Columns: 4 different datapoints. FFG: Fully-factorized Gaussian. Flow: Using a flexible approximate distribution. Amortized: Using amortized parameters. Optimal: Parameters optimized for individual datapoints. The green distributions are the true posterior distributions, highlighting the mismatch with the approximation.

Even though it is clear that failed inference would lead to a failed generative model, little quantitative assessment has been done showing the effect of the approximate posterior on the true posterior. Burda et al. (2016) visually demonstrate that when trained with an importance-weighted approximate posterior, the resulting true posterior is more complex than those trained with fully-factorized Gaussian approximations. We extend this observation quantitatively in the setting of flow-based approximate inference.

## 5 EXPERIMENTAL RESULTS

### 5.1 INTUITION THROUGH VISUALIZATION

To begin, we would like to gain some insight into the properties of inference in VAEs by visualizing different distributions in the latent space. To this end, we trained a VAE with a two-dimensional latent space on MNIST. We show contour plots of various distributions in the latent space in Fig. 2. The first row contains contour plots of the true posteriors $p(z|x)$ for four different training datapoints (columns). We have selected these four examples to highlight different inference phenomena. The amortized FFG row refers to the output of the recognition net, in this case, a fully-factorized Gaussian (FFG) approximation. Optimal FFG is the FFG that best fits the posterior of the datapoint. Optimal Flow is the optimal fit of a flexible distribution to the same posterior, where the flexible distribution we use is described in Section 3.2.

Posterior A is an example of a distribution where FFG can fit well. Posterior B is an example of dependence between dimensions, demonstrating the limitation of having a factorized approximation. Posterior C highlights a shortcoming of performing amortization with a limited-capacity recognition network, where the amortized FFG shares little support with the true posterior. Posterior D is a bimodal distribution which demonstrates the ability of the flexible approximation to fit to complex distributions, in contrast to the simple FFG approximation. These observations raise the following question: in more typical VAEs, is the amortization of inference the leading cause of the distribution mismatch, or is it the choice of approximation?

| | MNIST | | Fashion-MNIST | | CIFAR-10 | |
| --- | --- | --- | --- | --- | --- | --- |
| | $q_{FFG}$ | $q_{Flow}$ | $q_{FFG}$ | $q_{Flow}$ | $q_{FFG}$ | $q_{Flow}$ |
| $\log \hat{p}(x)$ | -89.80 | -88.94 | -97.47 | -97.41 | -14913.15 | -14914.45 |
| $\mathcal{L}_{\text{VAE}}[q_{Flow}^*]$ | -90.80 | -90.38 | -98.92 | -99.10 | -14914.22 | -14915.57 |
| $\mathcal{L}_{\text{VAE}}[q_{FFG}^*]$ | -91.23 | -113.54 | -100.53 | -132.46 | -14915.40 | -14919.08 |
| $\mathcal{L}_{\text{VAE}}[q]$ | -92.57 | -91.79 | -104.75 | -103.76 | -14976.57 | -14975.12 |
| Approximation | 1.43 | 1.44 | 3.06 | 1.69 | 2.25 | 1.12 |
| Amortization | 1.34 | 1.41 | 4.22 | 4.66 | 61.17 | 59.55 |
| Inference | 2.77 | 2.85 | 7.28 | 6.35 | 63.42 | 60.67 |

Table 2: Inference Gaps. The columns $q_{FFG}$ and $q_{Flow}$ refer to the variational distribution used for training the model. All numbers are in nats.

## 5.2 AMORTIZATION VS APPROXIMATION GAP

Here we will compare the influence that the approximation and amortization errors have on the total inference gap. Table 2 are results from training on MNIST, Fashion-MNIST and CIFAR-10. For each dataset, we trained two different approximate posterior distributions: a fully-factorized Gaussian, $q_{FFG}$, and a flexible distribution, $q_{Flow}$. Due to the computational cost of optimizing the local parameters for each datapoint, our evaluation is performed on a subset of 1000 datapoints for MNIST and Fashion-MNIST and a subset of 100 datapoints for CIFAR-10.

For MNIST, we see that the amortization and approximation gaps each account for nearly half of the inference gap. On Fashion-MNIST, which is a more difficult dataset to model, the amortization gap becomes larger than the approximation gap. Similarly for CIFAR-10, we see that the amortization gap is much more significant than the approximation gap. Thus, for the three datasets and model architectures that we tested, the amortization gap seems to be the prominent cause of inference suboptimality, especially when the difficulty of the dataset increases. This analysis indicates that improvements in inference will likely be a result of reducing amortization error, rather than approximation errors.

With these results in mind, would simply increasing the capacity of the encoder improve the amortization gap? We examined this by training the MNIST and Fashion-MNIST models from above but with larger encoders. See Section 6.1.2 for implementation details. Table 3 are the results of this experiment. Comparing to Table 2, we see that for both datasets and both variational distributions, the inference gap decreases and the decrease is mainly due to a reduction in the amortization gap.

| | MNIST | | Fashion-MNIST | |
| --- | --- | --- | --- | --- |
| | $q_{FFG}$ | $q_{Flow}$ | $q_{FFG}$ | $q_{Flow}$ |
| $\log \hat{p}(x)$ | -89.61 | -88.99 | -95.99 | -96.18 |
| $\mathcal{L}_{\text{VAE}}[q_{Flow}^*]$ | -90.65 | -90.44 | -97.40 | -97.91 |
| $\mathcal{L}_{\text{VAE}}[q_{FFG}^*]$ | -91.07 | -108.71 | -99.64 | -129.7 |
| $\mathcal{L}_{\text{VAE}}[q]$ | -92.18 | -91.19 | -102.73 | -101.67 |
| Approximation | 1.46 | 1.45 | 3.65 | 1.73 |
| Amortization | 1.11 | 0.75 | 3.09 | 3.76 |
| Inference | 2.56 | 2.20 | 6.74 | 5.49 |

Table 3: Larger Encoder. The columns $q_{FFG}$ and $q_{Flow}$ refer to the variational distribution used for training the model. All numbers are in nats.

### 5.2.1 INFLUENCE OF FLOWS ON AMORTIZATION GAP

The common reasoning for increasing the expressiveness of the approximate posterior is to minimize the difference between the true and approximate, i.e. reduce the approximation gap. However, given that the expressive approximation is often accompanied by many additional parameters, we would like to know if it has an influence on the amortization error.

To investigate this, we trained a VAE in the same manner as Section 5.2. After training, we kept the generator fixed and trained new encoders to fit to the fixed posterior. Specifically, we trained a small

encoder with a factorized Gaussian $q$ distribution to obtain a large amortization gap. We then trained a small encoder with a flow distribution. See Section 6.2 for the details of the experiment. The results are shown in Table 4. As expected, we observe that the small encoder has a very large amortization gap. However, when we use $q_{Flow}$ as the approximate distribution, we see the approximation gap decrease, but more importantly, there is a significant decrease in the amortization gap. This indicates that the parameters used for increasing the complexity of the approximation also play a large role in diminishing the amortization error.

| Variational Family | $q_{FFG}$ | $q_{Flow}$ |
|---|---|---|
| $\log \hat{p}(x)$ | -84.70 | -84.70 |
| $\mathcal{L}_{\text{VAE}}[q^*]$ | -86.61 | -85.48 |
| $\mathcal{L}_{\text{VAE}}[q]$ | -129.83 | -98.58 |
| Approximation | 1.91 | 0.78 |
| Amortization | 43.22 | 13.10 |
| Inference | 45.13 | 13.88 |

Table 4: Influence of Flows on the Amortization Gap. The parameters used to increase the flexibility of the approximate distribution also reduce the amortization gap. See Section 5.2.1 for details of the experiment.

These results are expected given that the parameterization of the Flow distribution can be interpreted as an instance of the RevNet (Gomez et al., 2017) which has demonstrated that Real-NVP like transformations (Dinh et al., 2017) can model complex functions similar to typical MLPs. Thus the flow transformations we employ should also be expected to increase the expressiveness while also increasing the capacity of the encoder. The implication of this observation is that models which improve the flexibility of their variational approximation, and attribute their improved results to the increased expressiveness, may have actually been due to the reduction in amortization error.

## 5.3 Influence of Approximate Posterior on True Posterior

We have seen that increasing the expressiveness of the approximation improves the marginal likelihood of the trained model, but to what amount does it alter the true posterior? Will a factorized Gaussian approximation cause the true posterior to be more like a factorized Gaussian or is the true posterior mostly fixed? Just as it is hard to evaluate a generative model by visually inspecting samples from the model, its hard to say how Gaussian the true posterior is by visual inspection. We can quantitatively determine how close the posterior is to a fully factorized Gaussian (FFG) distribution by comparing the marginal log-likelihood estimate, $\log \hat{p}(x)$, and the Optimal FFG bound, $\mathcal{L}_{\text{VAE}}[q^*_{FFG}]$. In other words, we are estimating the KL divergence between the optimal Gaussian and the true posterior, $\text{KL}(q^*(z|x)||p(z|x))$.

In Table 2 on MNIST, the Optimal Flow improves upon the Optimal FFG for the FFG trained model by 0.4 nats. In contrast, on the Flow trained model, the difference increases to 12.5 nats. This suggests that the true posterior of a FFG-trained model is closer to FFG than the true posterior of the Flow-trained model. The same observation can be made on the Fashion-MNIST dataset. This implies that the decoder can learn to have a true posterior that fits better to the approximation. Although the generative model can learn to have a posterior that fits to the approximation, it seems that not having this constraint, ie. using a flexible approximate, results in better generative models.

We can use these observations to help justify our approximation and amortization gap results of Section 5.2. Those results showed that the amortization error is often the main cause of inference suboptimality. One reason for this is that the generator accommodates to the choice of approximation, as shown above, thus reducing the approximation error.

Given that we have seen that the generator could accommodate to the choice of approximation, our next question is whether a generator with more capacity can accommodate more. To this end, we trained VAEs with decoders of different sizes and measured the approximation gaps. Specifically, we trained decoders with 0, 2, and 4 hidden layers on MNIST. See Table 5 for the results. We see that as the capacity of the decoder increases, the approximation gap decreases. This result implies that the more flexible the generator, the less flexible the approximate distribution needs to be.

| Generator Hidden Layers | 0 | 2 | 4 |
|---|---|---|---|
| $\log \hat{p}(x)$ | -100.52 | -86.61 | -83.82 |
| $\mathcal{L}_{\text{VAE}}[q_{FFG}^*]$ | -104.42 | -84.78 | -82.19 |
| Approximation Gap | 3.90 | 1.83 | 1.63 |

Table 5: Increased decoder capacity reduces approximation gap. All numbers are in nats.

### 5.3.1 ANNEALING THE ENTROPY

Typical warm-up (Bowman et al., 2015; Sønderby et al., 2016) refers to annealing KL $(q(z|x)||p(z))$ during training. This can also be interpreted as performing maximum likelihood estimation (MLE) early on during training. This optimization technique is known to help prevent the latent variable from degrading to the prior (Burda et al., 2016; Sønderby et al., 2016). We employ a similar annealing scheme during training. Rather than annealing the KL divergence, we anneal the entropy of the approximate distribution $q$:

$$\mathbb{E}_{z \sim q(z|x)} \left[ \log p(x,z) - \lambda \log q(z|x) \right],$$

where $\lambda$ is annealed from 0 to 1 over training. This can be interpreted as *maximum a posteriori* (MAP) in the initial phase. Due to its similarity, we will also refer to this technique as warm-up.

We find that warm-up techniques, such as annealing the entropy, are important for allowing the true posterior to be more complex. Table 6 are results from a model trained without the entropy annealing schedule. Comparing these results to Table 2, we observe that the difference between $\mathcal{L}_{\text{VAE}}[q_{FFG}^*]$ and $\mathcal{L}_{\text{VAE}}[q_{Flow}^*]$ is significantly smaller without entropy annealing. This indicates that the true posterior is more Gaussian when entropy annealing is not used. This suggests that, in addition to preventing the latent variable from degrading to the prior, entropy annealing allows the true posterior to better utilize the flexibility of the expressive approximation, resulting in a better trained model.

| | MNIST | | Fashion-MNIST | |
|---|---|---|---|---|
| | $q_{FFG}$ | $q_{Flow}$ | $q_{FFG}$ | $q_{Flow}$ |
| $\log \hat{p}(x)$ | -89.82 | -89.52 | -102.56 | -102.88 |
| $\mathcal{L}_{\text{VAE}}[q_{Flow}^*]$ | -90.96 | -90.45 | -103.73 | -104.02 |
| $\mathcal{L}_{\text{VAE}}[q_{FFG}^*]$ | -90.84 | -92.25 | -103.85 | -105.80 |
| $\mathcal{L}_{\text{VAE}}[q]$ | -92.33 | -91.75 | -106.90 | -107.01 |
| Approximation | 1.02 | 0.93 | 1.29 | 1.14 |
| Amortization | 1.49 | 1.30 | 3.05 | 2.29 |
| Inference | 2.51 | 2.23 | 4.34 | 4.13 |

Table 6: Models trained without entropy annealing. The columns $q_{FFG}$ and $q_{Flow}$ refer to the variational distribution used for training the model. All numbers are in nats.

## 6 CONCLUSION

In this paper, we investigated how encoder capacity, approximation choice, decoder capacity, and model optimization influence inference suboptimality in terms of the approximation and amortization gaps. We found that the amortization gap is often the leading source of inference suboptimality and that the generator reduces the approximation gap by learning a true posterior that fits to the choice of approximate distribution. We showed that the parameters used to increase the expressiveness of the approximation play a role in generalizing inference rather than simply improving the complexity of the approximation. We confirmed that increasing the capacity of the encoder reduces the amortization error. We also showed that optimization techniques, such as entropy annealing, help the generative model to better utilize the flexibility of the expressive variational distribution. Computing these gaps can be useful for guiding improvements to inference in VAEs. Future work includes evaluating other types of expressive approximations and more complex likelihood functions.

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

## APPENDIX

### 6.1 MODEL ARCHITECTURES AND TRAINING HYPERPARAMETERS

#### 6.1.1 2D VISUALIZATION

The VAE model of Fig. 2 uses a decoder $p(x|z)$ with architecture: $2 - 100 - 784$, and an encoder $q(z|x)$ with architecture: $784 - 100 - 4$. We use tanh activations and a batch size of 50. The model is trained for 3000 epochs with a learning rate of $10^{-4}$ using the ADAM optimizer (Kingma & Ba, 2014).

#### 6.1.2 MNIST & FASHION-MNIST

Both MNIST and Fashion-MNIST consist of a training and test set with 60k and 10k datapoints respectively, where each datapoint is a 28x28 grey-scale image. We rescale the original images so that pixel values are within the range $[0, 1]$. For MNIST, We use the statically binarized version described by Larochelle & Bengio (2008). We also binarize Fashion-MINST *statically*. For both datasets, we adopt the Bernoulli likelihood for the generator.

The VAE models for MNIST and Fashion-MNIST experiments have the same architecture given in table 7. The flow configuration is given in table 8.

| Inference Network | Generator |
|:---:|:---:|
| Input $\in \mathbb{R}^{784}$ | Input $\in \mathbb{R}^{50}$ |
| FC. 200-ELU-FC. 200-ELU-FC. 50+50 | FC. 200-ELU-FC. 200-ELU-FC. 784-Sigmoid |

Table 7: Neural net architecture for MNIST/Fashion-MNIST experiments.

In the large encoder setting, we change the number of hidden units for the inference network to be 500, instead of 200. The warm-up models are trained with a linear schedule over the first 400 epochs according to Section 5.3.1.

The activation function is chosen to be the exponential linear unit (ELU, Clevert et al. (2015)), as we observe improved performance compared to tanh. We follow the same learning rate schedule and train for the same amount of epochs as described by Burda et al. (2016). All models are trained with the a batch-size of 100 with ADAM.

### 6.1.3   CIFAR-10

CIFAR-10 consists of a training and test dataset with 50k and 10k datapoints respectively, where each datapoint is a $32 \times 32$ color image. We rescale individual pixel values to be in the range $[0, 1]$. We follow the *discretized logistic* likelihood model adopted by Kingma et al. (2016), where each input channel has its own scale learned by an MLP. For the latent variable, we use a 32-dimensional factorized Gaussian for $q(z|x)$ following Kingma et al. (2016). For all neural networks, ELU is chosen to be the activation function. The specific network architecture is shown in Table 9.

We adopt a gradually decreasing learning rate with an initialize value of $10^{-3}$. Warm-up is applied with a linear schedule over the first 20 epochs. All models are trained with a batch-size of 100 with ADAM. Early-stopping is applied based on the performance on the held-out set.

For the model with expressive inference, we use four flow steps as opposed to only two in MNIST/Fashion-MNIST experiments.

### 6.2   INFLUENCE OF FLOWS ON AMORTIZATION GAP EXPERIMENT

The aim of this experiment is to show that the parameters used for increasing the expressiveness of the approximation also contribute to reducing the amortization error. To show this, we train a VAE on MNIST, discard the encoder, then retrain two encoders on the fixed decoder: one with a factorized Gaussian distribution and the other with a parameterized 'flow' distribution. We use fixed decoder so that the true posterior is constant for both encoders. See 5.2.1 for the results and below for the architecture details.

The architecture of the decoder is: $D_Z - 200 - 200 - D_X$. The architecture of the encoder used to train the decoder is $D_X - 200 - 200 - 2D_Z$. The approximate distribution $q(z|x)$ is a factorized Gaussian.

Next, we describe the encoders which were trained on the fixed trained decoder. In order to highlight a large amortization gap, we employed a very small encoder architecture: $D_X - 2D_Z$. This encoder has no hidden layers, which greatly impoverishes its ability and results in a large amortization gap.

We compare two approximate distributions $q(z|x)$. Firstly, we experiment with the typical fully factorized Gaussian (FFG). The second is what we call a flow distribution. Specifically, we use the transformations of Dinh et al. (2017). We also include an auxiliary variable so we don't need to select how to divide the latent space for the transformations. The approximate distribution over the latent $z$ and auxiliary variable $v$ factorizes as: $q(z, v|x) = q(z|x)q(v)$. The $q(v)$ distribution is simply a N(0,1) distribution. Since we're using a auxiliary variable, we also require the $r(v|z)$ distribution which we parameterize as $r(v|z)$: $[D_Z] - 50 - 50 - 2D_Z$. The flow transformation is the same as in Section 3.2, which we apply twice.

| $q(v_0|z_0)$ | $r(v_T|z_T)$ |
|:---:|:---:|
| Input $\in \mathbb{R}^{50}$ | Input $\in \mathbb{R}^{50}$ |
| FC. 100-ELU-FC. 100-ELU-FC. 50+50 | FC. 100-ELU-FC. 100-ELU-FC. 50+50 |

| $q(v_{t+1}, z_{t+1}|v_t, z_t)$ | |
|:---:|:---:|
| $\sigma_1(\cdot), \sigma_2(\cdot)$ | $\mu_1(\cdot), \mu_2(\cdot)$ |
| Input $\in \mathbb{R}^{50}$ | Input $\in \mathbb{R}^{50}$ |
| FC. 100-ELU-FC. 100-ELU-FC. 50 | FC. 100-ELU-FC. 100-ELU-FC. 50 |

Table 8: Flow setting for MNIST/Fashion-MNIST experiments. $q(v_T, z_T|v_0, z_0)$ consists of two normalizing flows given in the second tabular.

| Inference Network | Generator |
|---|---|
| Input $32 \times 32$ color image | Input $\in \mathbb{R}^{32}$ |
| $4 \times 4$ conv. 64 channels. stride 2. BN | FC. $256 \times 2 \times 2$ ELU; FC. 64-ELU-FC. 32-ELU-FC. 3 |
| $4 \times 4$ conv. 128 channels. stride 2. BN | $4 \times 4$ deconv. 128 channels. stride 2. BN |
| $4 \times 4$ conv. 256 channels. stride 2. BN | $4 \times 4$ deconv. 64 channels. stride 2. BN |
| FC. $32 + 32$. output layer for mean and log-variance | $4 \times 4$ deconv. 3 channels. stride 2. Sigmoid |

Table 9: Network architecture for CIFAR-10 experiments. For the generator, one of the MLPs immediately after the input layer of the generator outputs channel-wise scales for the discretized logistic likelihood model. BN stands for batch-normalization.

### 6.3 COMPUTATION OF THE DETERMINANT FOR FLOW

The overall mapping $f$ that performs $(z, v) \mapsto (z', v')$ is the composition of two sheer mappings $f_1$ and $f_2$ that respectively perform $(z, v) \mapsto (z, v')$ and $(z, v') \mapsto (z', v')$. Since the Jacobian of either one of the sheer mappings is diagonal, the determinant of the composed transformation's Jacobian $Df$ can be easily computed:

$$\det(Df) = \det(Df_1)\det(Df_2) = \Big(\prod_{i=1}^{n} \sigma_1(z)_i\Big)\Big(\prod_{j=1}^{n} \sigma_2(v')_j\Big).$$

### 6.4 LOCAL OPTIMIZATION OF APPROXIMATE DISTRIBUTION

For the local FFG optimization, we initialize the mean and variance as the prior, i.e. $\mathcal{N}(0, I)$. We optimize the mean and variance using the Adam optimizer with a learning rate of $10^{-3}$. To determine convergence, after every 100 optimization steps, we compute the average of the previous 100 ELBO values and compare it to the best achieved average. If it does not improve for 10 consecutive iterations then the optimization is terminated. For the Flow model, the same process is used to optimize all of its parameters. All neural nets for the flow were initialized with a variant of the Xavier initilization (Glorot & Bengio, 2010). We use 100 Monte Carlo samples to compute the ELBO to reduce variance.

### 6.5 ANNEALED IMPORTANCE SAMPLING

Annealed importance sampling (AIS, Neal (2001); Jarzynski (1997)) is a means of computing a lower bound to the marginal log-likelihood. Similarly to the importance weighted bound, AIS must sample a proposal distribution $f_1(z)$ and compute the density of these samples, however, AIS then transforms the samples through a sequence of reversible transitions $\mathcal{T}_t(z'|z)$. The transitions anneal the proposal distribution to the desired distribution $f_T(z)$.

Specifically, AIS samples an initial state $z_1 \sim f_1(z)$ and sets an initial weight $w_1 = 1$. For the following annealing steps, $z_t$ is sampled from $\mathcal{T}_t(z'|z)$ and the weight is updated according to:

$$w_t = w_{t-1} \frac{f_t(z_{t-1})}{f_{t-1}(z_{t-1})}.$$

This procedure produces weight $w_T$ such that $\mathbb{E}[w_T] = \mathcal{Z}_T / \mathcal{Z}_1$, where $Z_T$ and $Z_1$ are the normalizing constants of $f_T(z)$ and $f_1(z)$ respectively. This pertains to estimating the marginal likelihood when the target distribution is $p(x, z)$ when we integrate with respect to $z$.

Typically, the intermediate distributions are simply defined to be geometric averages: $f_t(z) = f_1(z)^{1-\beta_t} f_T(z)^{\beta_t}$, where $\beta_t$ is monotonically increasing with $\beta_1 = 0$ and $\beta_T = 1$. When $f_1(z) = p(z)$ and $f_T(z) = p(x, z)$, the intermediate distributions are: $f_i(x) = p(z)p(x|z)^{\beta_i}$.

Model evaluation with AIS appears early on in the setting of deep belief networks (Salakhutdinov & Murray, 2008). AIS for decoder-based models was also used by Wu et al. (2017). They validated the accuracy of the approach with Bidirectional Monte Carlo (BDMC, Grosse et al. (2015)) and demonstrated the advantage of using AIS over the IWAE bound for evaluation when the inference network overfits to the training data.

## 6.6    THE INFERENCE GAP

How well is inference done in VAEs during training? Are we close to doing the optimal or is there much room for improvement? To answer this question, we quantitatively measure the inference gap: the gap between the true marginal log-likelihood and the lower bound. This amounts to measuring how well inference is being done during training. Since we cannot compute the exact marginal log-likelihood, we estimate it using the maximum of any of its lower bounds, described in 3.3.

Fig. 3a shows training curves for a FFG and Flow inference network as measured by the VAE, IWAE, and AIS bounds on the training and test set. The inference gap on the training set with the FFG model is 3.01 nats, whereas the Flow model is 2.71 nats. Accordingly, Fig. 3a shows that the training IWAE bound is slightly tighter for the Flow model compared to the FFG. Due to this lower inference gap during training, the Flow model achieves a higher AIS bound on the test set than the FFG model.

To demonstrate that a very small inference gap can be achieved, even with a limited approximation such as a factorized Gaussian, we train the model on a small dataset. In this experiment, our training set consists of 1000 datapoints randomly chosen from the original MNIST training set. The training curves on this small datatset are show in Fig. 3b. Even with a factorized Gaussian distribution, the inference gap is very small: the AIS and IWAE bounds are overlapping and the VAE is just slightly below. Yet, the model is overfitting as seen by the decreasing test set bounds.

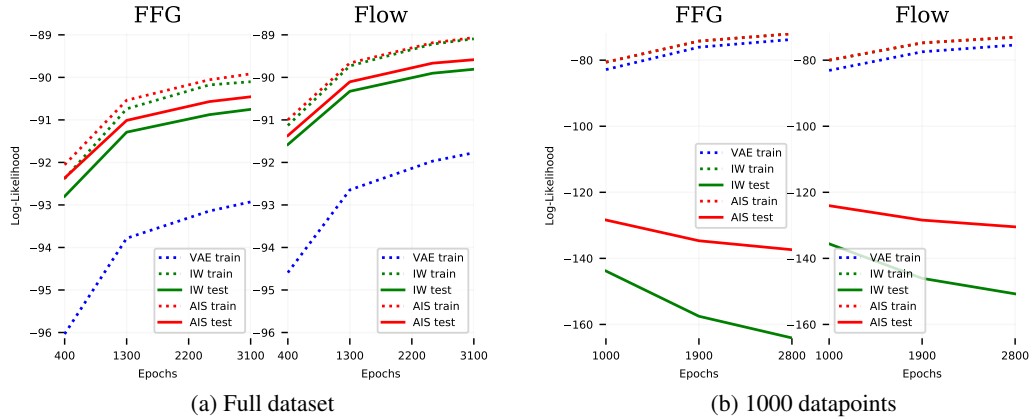

(a) Full dataset                               (b) 1000 datapoints

Figure 3: Training curves for a FFG and a Flow inference model on MNIST. AIS provides the tightest lower bound and is independent of encoder overfitting. There is little difference between FFG and Flow models trained on the 1000 datapoints since inference is nearly equivalent.

### 6.6.1    ENCODER AND DECODER OVERFITTING

We will begin by explaining how we separate encoder from decoder overfitting. Decoder overfitting is the same as in the regular supervised learning scenario, where we compare the train and test error. To measure decoder overfitting independently from encoder overfitting, we use the AIS bound since it is encoder-independent. Thus we can observe decoder overfitting through the AIS test training curve. In contrast, the encoder can only overfit in the sense that the recognition network becomes unsuitable for computing the marginal likelihood on the test set. Thus, encoder overfitting is computed by: $\mathcal{L}_{\text{AIS}} - \mathcal{L}_{\text{IW}}$ on the test set.

For the small dataset of Fig. 3b, it clear that there is significant encoder and decoder overfitting. A model trained in this setting would benefit from regularization. For Fig. 3a, the model is not overfit and would benefit from more training. However, there is some encoder overfitting due to the gap between the AIS and IWAE bounds on the test set. Comparing the FFG and Flow models, it appears that the Flow does not have a large effect on encoder or decoder overfitting.

## 6.7 GAUSSIAN LATENTS WITH FULL COVARIANCE

The flexiblity of the Gaussian family with arbitrary covariance lies between that of FFG and Flow. With covariance, the Gaussian distribution can model interactions between different latent dimensions. Yet, compared to Flow, its expressiveness is limited due to its inability to model higher order interactions and its unimodal nature.

To apply the reparameterization trick, we perform the Cholesky decomposition on the covariance matrix: $\Sigma = LL^\top$, where $L$ is lower triangular. A sample from $\mathcal{N}(\mu, \Sigma)$ could be obtained by first sampling from a unit Gaussian $\epsilon \sim \mathcal{N}(0, I)$, then computing $z = \mu + L\epsilon$.

To analyze the capability of the Gaussian family, we train several VAEs on MNIST and Fashion-MNIST with the approximate posterior $q(z|x)$ being a Gaussian with full covariance. To inspect how well inference is done, we perform the local optimizations described in Section 5.2 with FFG and Flow.

|  | MNIST | Fashion-MNIST |
| --- | --- | --- |
| $\log \hat{p}(x)$ | -89.28 | -96.46 |
| $\mathcal{L}_{\text{VAE}}[q^*_{Flow}]$ | -90.69 | -98.19 |
| $\mathcal{L}_{\text{VAE}}[q^*_{FFG}]$ | -101.84 | -107.89 |
| $\mathcal{L}_{\text{VAE}}[q]$ | -92.05 | -102.93 |

Table 10: Gaussian latents trained with full covariance.

We can see from table 10 that local optimization with FFG on a model trained with full covariance inference produces a bad lower bound. This resonates with the argument that the approximation has a significant influence on the true posterior as described in section 5.3.

Comparing to numbers in table 2, we can see that the full-covariance VAE trained on MNIST is nearly on par with that trained with Flow (-89.28 vs -88.94). For Fashion-MNIST, the full-covariance VAE even performs better by a large margin in terms of the estimated log-likelihood (-96.46 vs -97.41).

