# OpenReview forum: "Inference Suboptimality in Variational Autoencoders"
_ICLR.cc/2018/Conference — Invite to Workshop Track_

### Official Review · AnonReviewer3 · 2017-11-27
**new way of analyzing slack in ELBO for VAEs, contribution is limited**

**Rating:** 6
**Confidence:** 5

**Review:**

* EDIT: Increased score from 5 to 6 to reflect improvements made in the revision.

The authors break down the "inference gap" in VAEs (the slack in the variational lower bound) into two components: 1. the "amortization gap", measuring what part of the slack is due to amortizing inference using a neural net encoder, as compared to separate optimization per example. 2. the "approximation gap": the part of the slack due to using a restricted parametric form for the posterior approximation. They perform various experiments to analyze how these quantities depend on modeling decisions and data sets.

Breaking down the inference gap into its components is an interesting idea and could potentially provide insights when analyzing VAE performance and for further improving VAEs. I enjoyed reading the paper, but I think its contribution is on the small side for a conference paper. It would be a good workshop paper. The main limitation of the proposed method of analysis I think is that the two parts of the inference gap are not really separable: Because the VAE encoder is trained jointly with the decoder, the different limitations of the encoder and decoder all interact. E.g. one could imagine cases where jointly training the VAE encoder and decoder finds a local optimum where inference is perfect, but which is still much worse than the optimum that could be achieved if the encoder would have been more flexible. The authors do seem to realize this and they provide experiments examining this interaction. I think these experiments should be elaborated on. For example: What happens when the decoder is trained separately using more flexible inference (e.g. Hamiltonian MC) and the encoder is trained later? What happens when the encoder is optimized separately for each data point during training as well as testing?

---

> ### Author Response · Authors · 2018-01-04
> **Response to Reviewer 3**
>
>
> We would like to thank Reviewer 3 for providing a detailed review and interesting suggestions for further experimentation.
>
> Overall, we acknowledge that the different limitations of the encoder and decoder all interact, e.g. in Section 5.3 we have quantitatively demonstrated that a VAE trained with a factorized Gaussian, typically have a true posterior that is more like a factorized Gaussian. Without doubt, we also agree that there could be cases where the generative model fits embarrassingly to the data, and yet inference is perfect. However, this does not hinder our analysis of the gaps according to our definitions (Section 3.1). We would also like to note that, although our calculations of the gaps are only estimates, such a amortization-approximation decomposition may be valuable to guiding improvements to approximate inference.
>
> "What happens when the decoder is trained separately using more flexible inference (e.g. Hamiltonian MC) and the encoder is trained later? What happens when the encoder is optimized separately for each data point during training as well as testing?"
>
> These are interesting ideas. We performed local optimization of the variational parameters only for evaluation purposes. The quality of inference is an important factor for the optimization of the generator. Consequently, training with HMC would likely result in a better trained generator compared to training via amortized inference, especially early during training. We refer to [1] for experiments that explore optimizing the variational parameters in the inner loop of amortized inference during training.
>
> [1] R. G. Krishnan, D. Liang, and M. Hoffman.  On the challenges of learning with inference networks on sparse, high-dimensional data.ArXiv e-prints, October 2017

---

### Official Review · AnonReviewer2 · 2017-11-27
**This paper studies and attempts to break down the sources of errors in doing inference in VAEs on MNIST.**

**Rating:** 6
**Confidence:** 4

**Review:**

This paper studies the amortization gap in VAEs. Inference networks, in general, have two sources of approximation errors. One due to the function family of variational posterior distributions used in inference and the other due to choosing to amortize inference rather than doing per-data-point inference as in SVI.

They consider learning VAEs using two different choices of inference networks with (1) fully factorized Gaussian and (2) normalizing flows. The former is the de-facto choice of variational approximation used in VAEs and the latter is capable of expressing complex multi-modal distributions.

The inference gap is log p(x) - L[q], the approximation gap is log p(x) - L[q^* ] and the amortization gap is L[q^* ] - L[q]. The amortization gap is easily evaluated. To evaluate the first two, the authors use estimates (lower bounds of log p(x)) given from annealed importance sampling and the importance sampling based IWAE bound (the tighter of the two is used).

There are several different observations made via experiments in this work but one of the more interesting ones is quantifying that a deep generative model, when trained with a fully factorized gaussian posterior, realizes a true posterior distribution that is (more) approximately Gaussian. While this might be (known) intuition that people rely on when learning deep generative models, it is important to be able to test it, as this paper does. The authors study several discrete questions about the aforementioned inference gaps and how they vary on MNIST and FashionMNIST. The concerns I have about this work revolve around their choice of two small datasets and how much their results are affected by variance in the estimators.

Questions:
* How did you optimize the variational parameters for q^* and the flow parameters in terms of learning rate, stopping criteria etc.
* In Section 5.2, what is "strong inference"? This is not defined previously.
* Have you evaluated on a larger dataset such as CIFAR? FashionMNIST and MNIST are similar in many ways.
* Which kind of error would using a convolution architecture for the encoder decrease? Do you have insights on the role played by the architecture of the inference network and generative model?

I have two specific concerns:
* Did you perform any checks to verify whether the variance in the estimators use to bound log p(x) is controlled (for the specific # samples you use)? I'm concerned since the evaluation is only done on 100 points.
* In Section 5.2.1, L_iw is used to characterize encoder overfitting where the argument is that L_ais is not a function of the encoder, but L_iw is, and so the difference between the two summarizes how much the inference network has overfit. How is L_iw affected by the number of samples used in the estimator? Presumably this statement needs to be made while also keeping mind the number of importance samples. For example, if I increase the number of importance samples, even if I'm overfitting in Fig 3(b), wouldn't the green line move towards the red simply because my estimator depends less on a poor q?

Overall, I think this paper is interesting and presents a quantitative analysis of where the errors accrue due to learning with inference networks. The work can be made stronger by addressing some of the questions above such as what role is played by the neural architecture and whether the results hold up under evaluation on a larger dataset.

---

> ### Author Response · Authors · 2018-01-04
> **Response to Reviewer 2**
>
>
> We would like to thank Reviewer 2 for their thorough analysis of our work. We acknowledge their concerns and address their comments below:
>
> “How did you optimize the variational parameters for q^* and the flow parameters in terms of learning rate, stopping criteria etc.”
>
> Thank you for asking. We have added the description in section 6.4 of the Appendix.
>
> "What is "strong inference"? This is not defined previously."
>
> By strong inference, we mean there is a small inference gap. We understand that this could be confusing given no prior explanation, thus we’ve changed the wording accordingly.
>
> “Have you evaluated on a larger dataset such as CIFAR? FashionMNIST and MNIST are similar in many ways.”
>
> We acknowledge that both MNIST and Fashion-MNIST are similar datasets. To enhance our analysis, we performed some new experiments on CIFAR-10 whose result is added to Table 2 and analyzed in section 5.2.
>
> “Which kind of error would using a convolution architecture for the encoder decrease?"
>
> Although we have not experimented extensively on the influence of the encoder architectures, more powerful encoders usually lead to lower amortization error. Our experiments with larger encoders demonstrates this (Section 5.2).
>
> "Do you have insights on the role played by the architecture of the inference network and generative model?”
>
> Thank you for the interesting question. Our most recent draft contains new results exploring the effect of increasing the capacity of the generative model. We observe that increasing the capacity leads to true posteriors that fit better to the choice of approximation. (Section 5.3)
>
> “Did you perform any checks to verify whether the variance in the estimators use to bound log p(x) is controlled (for the specific # samples you use)? I'm concerned since the evaluation is only done on 100 points.”
>
> We acknowledge that the variance of the bounds can be quite large, and the numbers we obtained for evaluating 100 datapoints might suffer from this. Thus, we re-performed all experiments on MNIST and Fashion-MNIST with 1k datapoints to reduce the variance. The new results are consistent with our previous results.
>
> "Presumably this statement needs to be made while also keeping mind the number of importance samples."
>
> Thank you for pointing this out. Yes, this statement needs to be made while also keeping in mind the number of importance samples, since measuring the overfitting is dependent on the number of samples.

---

### Official Review · AnonReviewer1 · 2017-11-27
**An interesting topic, and some interesting results, but probably a bit below the bar.**

**Rating:** 6
**Confidence:** 5

**Review:**

=======
Update:

The new version addresses some of my concerns. I think this paper is still pretty borderline, but I increased my rating to a 6.
=======

This article examines the two sources of loose bounds in variational autoencoders, which the authors term “approximation error” (slack due to using a limited variational family) and “amortization error” (slack due to the inference network not finding the optimal member of that family).

The existence of amortization error is often ignored in the literature, but (as the authors point out) it is not negligible. It has been pointed out before in various ways, however:
* Hjelm et al. (2015; https://arxiv.org/pdf/1511.06382.pdf) observe it for directed belief networks (admittedly a different model class).
* The ladder VAE paper by Sonderby et al. (2016, https://arxiv.org/pdf/1602.02282.pdf) uses an architecture that reduces the work that the encoder network needs to do, without increasing the expressiveness of the variational approximation. That this approach works well implies that amortization error cannot be ignored.
* The structured VAE paper by Johnson et al. (2016, https://arxiv.org/abs/1603.06277) also proposes an architecture that reduces the load on the inference network.
* The very recent paper by Krishnan et al. (posted to arXiv days before the ICLR deadline, although a workshop version was presented at the NIPS AABI workshop last year; http://approximateinference.org/2016/accepted/KrishnanHoffman2016.pdf) examines amortization error as a core cause of training failures in VAEs. They also observe that the gap persists at test time, although it does not examine how it relates to approximation error.

Since these earlier results existed, and approximation-amortization decomposition is fairly simple (although important!), the main contributions of this paper are the empirical studies. I will try to summarize the main novel (i.e., not present elsewhere in the literature) results of these:

Section 5.1:
Inference networks with FFG approximations can produce qualitatively embarrassing approximations.

Section 5.2:
When trained on a small dataset, training amortization error becomes negligible. I found this surprising, since it’s not at all clear why dataset size should lead to “strong inference”. It seems like a more likely explanation is that the decoder doesn’t have to work as hard to memorize the training set, so it has some extra freedom to make the true posterior look more like a FFG.

Also, I think it’s a bit of an exaggeration to call a gap of 2.71 nats “much tighter” than a gap of 3.01 nats.

Section 5.3:
Amortization error is an important contributor to the slack in the ELBO on MNIST, and the dominant contributor on the more complicated Fashion MNIST dataset. (This is totally consistent with Krishnan et al.’s finding that eliminating amortization error gave a bigger improvement for more complex datasets than for MNIST.)

Section 5.4:
Using a restricted variational family causes the decoder to learn to induce posteriors that are easier to approximate with that variational family. This idea has been around for a long time (although I’m having a hard time coming up with a reference).

These results are interesting, but given the empirical nature of this paper I would have liked to see results on more interesting datasets (Celeb-A, CIFAR-10, really anything but MNIST). Also, it seems as though none of the full-dataset MNIST models have been trained to convergence, which makes it a bit difficult to interpret some results.


A few more specific comments:

2.2.1: The \cdot seems extraneous to me.

5.1: What dataset/model was this experiment done on?

Figure 3: This can be inferred from the text (I think), but I had to remind myself that “IW train” and “IW test” refer only to the evaluation procedure, not the training procedure. It might be good to emphasize that you don’t train on the IWAE bound in any experiments.

Table 2: It would be good to see standard errors on these numbers; they may be quite high given that they’re only evaluated on 100 examples.

“We can quantitatively determine how close the posterior is to a FFG distribution by comparing the Optimal FFG bound and the Optimal Flow bound.”: Why not just compare the optimal with the AIS evaluation? If you trust the AIS estimate, then the result will be the actual KL divergence between the FFG and the true posterior.

---

> ### Author Response · Authors · 2018-01-04
> **Response to Reviewer 1**
>
>
> We would like to thank Reviewer 1 for their detailed comments regarding our contributions and providing citations of relevant work.
>
> We address their comments below:
>
> “When trained on a small dataset, training amortization error becomes negligible. I found this surprising, since it’s not at all clear why dataset size should lead to 'strong inference' "
>
> We believe the explanation for better inference on a smaller dataset is mostly due to the encoder having fewer datapoints to memorize, reducing the amortization error. Our analysis with larger encoders in Section 5.2 is relevant to supporting this claim.
>
> "It seems like a more likely explanation is that the decoder doesn’t have to work as hard to memorize the training set, so it has some extra freedom to make the true posterior look more like a FFG."
>
> This idea is interesting. We believe that the decoder having to work less hard is related to reducing the approximation error. Our results of Section 5.3 explore this idea.
>
> "Also, I think it’s a bit of an exaggeration to call a gap of 2.71 nats “much tighter” than a gap of 3.01 nats."
>
> Yes, we agree. We have re-worded that statement.
>
> "Using a restricted variational family causes the decoder to learn to induce posteriors that are easier to approximate with that variational family."
>
> Yes, this idea has been around for a while. One example is demonstrated with visualizations in Appendix C of the IWAE paper, which we’ve noted in the Related Works section. Our results provide quantitative measurements of this intuition.
>
> “These results are interesting, but given the empirical nature of this paper I would have liked to see results on more interesting datasets (Celeb-A, CIFAR-10, really anything but MNIST). ”
>
> We agree that more extensive empirical results are important. To this end, we performed new experiments on CIFAR-10 whose results are added to Table 2 and section 5.2.
>
> “The \cdot seems extraneous to me.”
>
> Thank you for pointing it out, we have fixed this.
>
> "What dataset/model was this experiment done on?"
>
> We trained our VAE models on MNIST for the visualization.
>
> “It would be good to see standard errors on these numbers; they may be quite high given that they’re only evaluated on 100 examples.”
>
> We acknowledge that the variance of the bounds can be quite large, and the numbers we obtained for evaluating 100 datapoints might suffer from this. Thus, we re-performed all experiments on MNIST and Fashion-MNIST with 1k datapoints to reduce the variance. The new results are consistent with our previous results.
>
> “Why not just compare the optimal with the AIS evaluation? If you trust the AIS estimate, then the result will be the actual KL divergence between the FFG and the true posterior.”
>
> Thank you for pointing this out. We have corrected the analysis accordingly.

---

### Author Response · Authors · 2018-01-04
**Updated Paper**


We’d like to thank the reviewers for the thoughtful and thorough reviews.

The consensus of the reviews is that the contribution of the original paper was limited, of which we completely agree. We’ve thus taken steps to extend our results with relevant experiments. We’ve used the same methodology as before in settings that we think highlight and strengthen important points about the paper.

Main additions:

CIFAR-10: we’ve run the same experiments on the CIFAR-10 dataset in order to gain a more comprehensive view of inference suboptimality. (see Section 5.2 and Table 2)

More datapoints: previously we evaluated the various gaps on a subset of 100 datapoints. We’ve increased the subset to 1000 datapoints for most experiments in order to make our results more reliable. The new results are consistent with our previous results.

Influence of flows on amortization: we demonstrate that the parameters used in increasing the expressiveness of the approximate distribution also contribute to reducing the amortization error. (see section 5.2.1 and Table 4)

Influence of decoder capacity on approximation gap: we demonstrate that increasing the number of hidden layers of the decoder leads to smaller approximation gaps. (see Section 5.3 and Table 5)


Other modifications:

Title: We changed the title of the paper from ‘Inference Dissection in Variational Autoencoders’ to ‘Inference Suboptimality in Variational Autoencoders’ because we believe it better reflects the content of the paper.

Organization: We’ve moved some less relevant sections to the appendix, such as the description of AIS and our section on VAE under/overfitting.

Abstract: We’ve updated the abstract given the new contributions.

---

### Decision · Program_Chairs · 2018-01-29
**ICLR 2018 Conference Acceptance Decision**

**Decision:**

Invite to Workshop Track

**Comment:**

Thank you for submitting you paper to ICLR. This paper provides an informative analysis of the approximation contributions from the various assumptions made in variational auto-encoders. The revision has demonstrated the robustness of the paper’s conclusions, however these conclusions are arguably unsurprising. Although the work provides a thorough and interesting piece of detective work, the significance of the findings is not quite great enough to warrant publication.

Reviewer 1 was searching for a reference for work in similar vein to section 5.4: The second problem identified in the reference below shows examples where using an approximating distribution of a particular form biases the model parameter estimates to settings that mean the true posterior is closer to that form.

R. E. Turner and M. Sahani. (2011) Two problems with variational Expectation Maximisation for time-series models. Inference and Learning in Dynamic Models. Eds. D. Barber, T. Cemgil and S. Chiappa, Cambridge University Press, 104–123, 2011.